# A molecular filter for the cnidarian stinging response

Keiko Weir[1], Christophe Dupre[1], Lena van Giesen[1], Amy S-Y Lee[2], Nicholas W Bellono[1]*

[1]Department of Molecular and Cellular Biology, Harvard University, Cambridge, United States; [2]Department of Biology, Brandeis University, Waltham, United States

**Abstract** All animals detect and integrate diverse environmental signals to mediate behavior. Cnidarians, including jellyfish and sea anemones, both detect and capture prey using stinging cells called nematocytes which fire a venom-covered barb via an unknown triggering mechanism. Here, we show that nematocytes from *Nematostella vectensis* use a specialized voltage-gated calcium channel ($nCa_V$) to distinguish salient sensory cues and control the explosive discharge response. Adaptations in $nCa_V$ confer unusually sensitive, voltage-dependent inactivation to inhibit responses to non-prey signals, such as mechanical water turbulence. Prey-derived chemosensory signals are synaptically transmitted to acutely relieve $nCa_V$ inactivation, enabling mechanosensitive-triggered predatory attack. These findings reveal a molecular basis for the cnidarian stinging response and highlight general principles by which single proteins integrate diverse signals to elicit discrete animal behaviors.

## Introduction

Jellyfish, sea anemones, and hydrozoans of the Cnidarian phylum use specialized cells called cnidocytes to facilitate both sensation and secretion required for prey capture and defense (*Watson and Mire-Thibodeaux, 1994b*). Two major types of cnidocytes contribute to prey capture by the tentacles of the starlet sea anemone (*Nematostella vectensis, Figure 1A*): (1) spirocytes, anthozoan-specific cells that extrude a thread-like organelle to ensnare prey, and (2) nematocytes, pan-cnidarian cells which eject a single-use venom-covered barb to mediate stinging (*Babonis and Martindale, 2017*). Sensory cues from prey act on nematocytes to trigger the explosive discharge of a specialized organelle (nematocyst) at an acceleration of up to $5.41 \times 10^6$ g, among the fastest of any biological process (*Holstein and Tardent, 1984*; *Nüchter et al., 2006*; *Figure 1B*). The nematocyst can only be discharged once and therefore stinging represents an energetically expensive process that is likely tightly regulated (*Watson and Mire-Thibodeaux, 1994b*, *Babonis and Martindale, 2014*). Indeed, simultaneously presented chemical and mechanical (chemo-tactile) cues are required to elicit nematocyte discharge (*Pantin, 1942a*; *Watson and Hessinger, 1989*; *Watson and Hessinger, 1992*; *Anderson and Bouchard, 2009*). Electrical stimulation of nematocytes increases the probability of discharge in a calcium ($Ca^{2+}$)-dependent manner (*Anderson and Mckay, 1987*; *McKay and Anderson, 1988*; *Santoro and Salleo, 1991*; *Gitter et al., 1994*; *Watson and Hessinger, 1994a*; *Anderson and Bouchard, 2009*), but direct recordings from nematocytes are limited and thus mechanisms by which environmental signals control discharge are not well studied.

Here, we demonstrate that nematocytes from *Nematostella vectensis* use a specialized $Ca_V2.1$ voltage-gated calcium channel orthologue ($nCa_V$) to integrate dynamic voltage signals produced by distinct sensory stimuli. We show nematocytes are intrinsically mechanosensitive but $nCa_V$ exhibits unique voltage-dependent inactivation that basally inhibits cellular activity, thereby preventing responses to extraneous mechanical stimuli, such as background water turbulence. We further show that sensory neurons make synaptic contact with nematocytes, and the neurotransmitter

*For correspondence:
nbellono@harvard.edu

Competing interests: The authors declare that no competing interests exist.

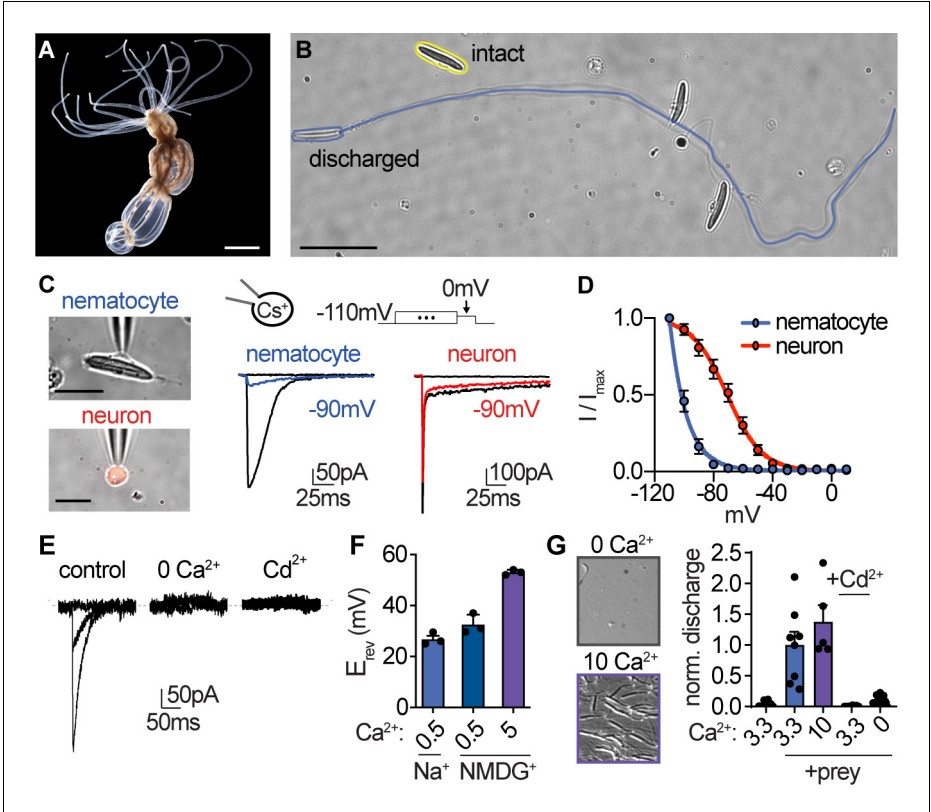

**Figure 1.** Nematocyte voltage-gated $Ca^{2+}$ currents exhibit sensitive voltage-dependent inactivation. (**A**) Starlet sea anemone (*Nematostella vectensis*). Scale bar = 3 mm. (**B**) Intact (yellow) and discharged nematocyte (blue). Scale bar = 20 μm. (**C**) *Left*: Representative patch clamp experiments from a nematocyte and tentacle neuron. Scale bar = 10 μm. *Right*: Nematocyte or neuron voltage-gated currents elicited by a maximally activating voltage pulse following 1 s pre-pulses to −110 mV (max current), −90 mV (colored), or 0 mV (inactivated, no current). (**D**) Nematocyte voltage-gated currents inactivated at very negative voltages compared with neurons. Nematocyte inactivation occurred at voltages more negative than could be measured compared with a sigmoidal inactivation relationship in neurons: nematocyte estimated $V_{i1/2}$ = -100.2 ± 0.4mV, n = 13 and neuron $V_{i1/2}$ = -70.8 ± 1.0mV, n = 9. Apparent activation thresholds were similar (*Figure 1—figure supplement 1A*). (**E**) Nematocyte voltage-gated currents elicited by −40 mV and 0 mV pulses were abolished in absence of external $Ca^{2+}$ and blocked by cadmium ($Cd^{2+}$). Representative of n = 4 for 0 $Ca^{2+}$ and three for $Cd^{2+}$, p<0.001 paired two-tailed student's t-test. (**F**) Nematocyte voltage-gated currents were $Ca^{2+}$-sensitive. Substitution of extracellular $Ca^{2+}$, but not $Na^+$ for $NMDG^+$, affected the reversal potential. n = 3–4, p<0.001 for 5 mM $Ca^{2+}$ versus other conditions, one-way ANOVA with post-hoc Tukey test. (**G**) Nematocyte discharge was minimal or absent in response to mechanical stimulation alone (n = 11, 3.3 mM $Ca^{2+}$). In the presence of prey extract, mechanically evoked discharge was similar in standard and higher concentration of extracellular $Ca^{2+}$ (n = 8 for 3.3 mM $Ca^{2+}$, n = 5 for 10 mM) and blocked by $Cd^{2+}$ (n = 8) or the removal of extracellular $Ca^{2+}$ (n = 15). Discharged nematocysts embedded in presented gelatin-coated coverslips were quantified. p<0.001 for + prey with 3.3 or 10 mM $Ca^{2+}$ versus other conditions, one-way ANOVA with post-hoc Bonferroni test. Data represented as mean ± sem.

The online version of this article includes the following source data and figure supplement(s) for figure 1:

**Source data 1.** Properties of nematocytes and neurons.
**Figure supplement 1.** Native nematocyte Ca$_V$ properties.

acetylcholine (ACh) elicits a hyperpolarizing response that relieves nCa$_V$ inactivation to allow for subsequent cellular stimulation and chemo-tactile-elicited discharge. Thus, we propose that the specialized voltage dependence of nCa$_V$ acts as a molecular filter for sensory discrimination.

## Results

### Nematocyte Ca$_V$ channels

We first obtained whole-cell patch clamp recordings from acutely dissociated nematocytes to investigate nematocyte signal transduction. Using intracellular cesium (Cs$^+$) to block potassium (K$^+$) currents revealed a voltage-gated inward current that was activated by positive or depolarized membrane voltages (I$_{CaV}$, *Figure 1C*). In response to sustained positive voltage, voltage-activated ion channels enter a non-conducting, inactivated state and cannot be activated until returned to a resting state by negative membrane voltage. This property generally serves to limit responses to repetitive or prolonged stimulation, similar to receptor desensitization. Remarkably, I$_{CaV}$ began to inactivate at voltages more negative than we could technically measure, thus demonstrating an unusual voltage sensitivity of this conductance (*Figure 1C,D*). To determine whether these properties were specific to nematocytes, we used a transgenic sea anemone with fluorescently labeled neurons to facilitate direct comparison between these excitable cell types (*Nakanishi et al., 2012*; *Figure 1C*). Neuronal voltage-gated currents had a lower threshold for activation and exhibited much weaker voltage-dependent inactivation (*Figure 1C,D*, *Figure 1—figure supplement 1A–D*), similar to currents found in neurons of other animals (*Hille, 2001*), indicating that nematocytes exhibit unusual voltage-dependent properties. Ion substitution and pore blocker experiments confirmed I$_{CaV}$ is a Ca$^{2+}$-sensitive current (*Figure 1E,F*), consistent with the contribution of extracellular Ca$^{2+}$ to chemo-tactile-induced discharge (*Watson and Hessinger, 1994a*; *Gitter et al., 1994*; *Figure 1G*). Increased concentrations of extracellular Ca$^{2+}$ did not affect inactivation of I$_{CaV}$ (*Figure 1—figure supplement 1E*), suggesting the enhanced voltage-dependent inactivation is intrinsic to the channel complex. This observation is important because it suggests I$_{CaV}$ renders nematocytes completely inactivated at typical resting membrane voltages and thus cells could not be stimulated from resting state.

To identify the ion channel mediating I$_{CaV}$, we generated a tentacle-specific transcriptome and aligned reads from nematocyte-enriched cells (*Sunagar et al., 2018*). This strategy allowed us to search for differentially expressed transcripts that might encode Ca$_V$ channel subunits (pore-forming α and auxiliary β and α2δ subunits). The orthologue of *cacnb2*, a β subunit of Ca$_V$ channels, was the highest expressed Ca$_V$ transcript in nematocyte-enriched cells, with levels 14-fold higher than other cells in the sea anemone (*Figure 2A*). β subunits can modulate voltage-dependence and trafficking in diverse ways depending on their splice isoform, interacting subunits, and cellular context (*Buraei and Yang, 2010*). Importantly, β subunits only interact with α subunits of high voltage-activated (HVA) calcium channels (*Perez-Reyes, 2003*). In agreement with robust β subunit expression, we found significant enrichment for *cacna1a*, the pore-forming subunit of HVA Ca$_V$2.1, and high expression of *cacna2d1* (*Figure 2—figure supplement 1A,B*). These observations are consistent with a previous report demonstrating specific expression of *cacna1a* in nematocytes of sea anemone tentacles and expression of β subunits in nematocytes from jellyfish (*Bouchard et al., 2006*; *Moran and Zakon, 2014*; *Bouchard and Anderson, 2014*). Expression of *cacna1h*, which does not interact with auxiliary subunits (*Buraei and Yang, 2010*), was also observed, albeit at lower levels and across all cells (*Figure 2—figure supplement 1A*). Thus, it remains possible that voltage-gated currents in nematocytes are not carried exclusively by one Ca$_V$ subtype.

Heterologous expression of *Nematostella* Ca$_V$ (nCa$_V$: *cacna1a*, *cacnb2*, and *cacna2d1*) produced voltage-gated currents with an apparent activation threshold nearly identical to the Ca$_V$ complex made from respective mammalian orthologues (mCa$_V$, *Figure 2B*, *Figure 2—figure supplement 1C, D*). Both channels had similar activation kinetics, but fast inactivation was significantly pronounced in nCa$_V$, resembling native I$_{CaV}$ (*Figure 2E*, *Figure 2—figure supplement 1E*). Importantly, nCa$_V$ voltage-dependent inactivation was greatly enhanced compared with mCa$_V$, regardless of the charge carrier (*Figure 2B*, *Figure 2—figure supplement 1F,G*). Similar to I$_{CaV}$, nCa$_V$ exhibited unusually-sensitive voltage-dependence and began to inactivate at voltages more negative than we could measure with an estimated midpoint inactivation voltage (V$_{i1/2}$)~80 mV more negative than mCa$_V$ (*Figure 2B*). Even with a holding potential of −70 mV, nCa$_V$ exhibited slow inactivation resulting in a drastic decrease in responses to depolarizing stimuli over time (*Figure 2C*). This slow inactivation was largely prevented by adjusting the holding potential to −110 mV, suggesting inactivation occurs when channels are in a closed-state at potentials near or more negative than typical resting membrane potential (*Figure 2—figure supplement 1H*). Importantly, nCa$_V$ rapidly recovered from

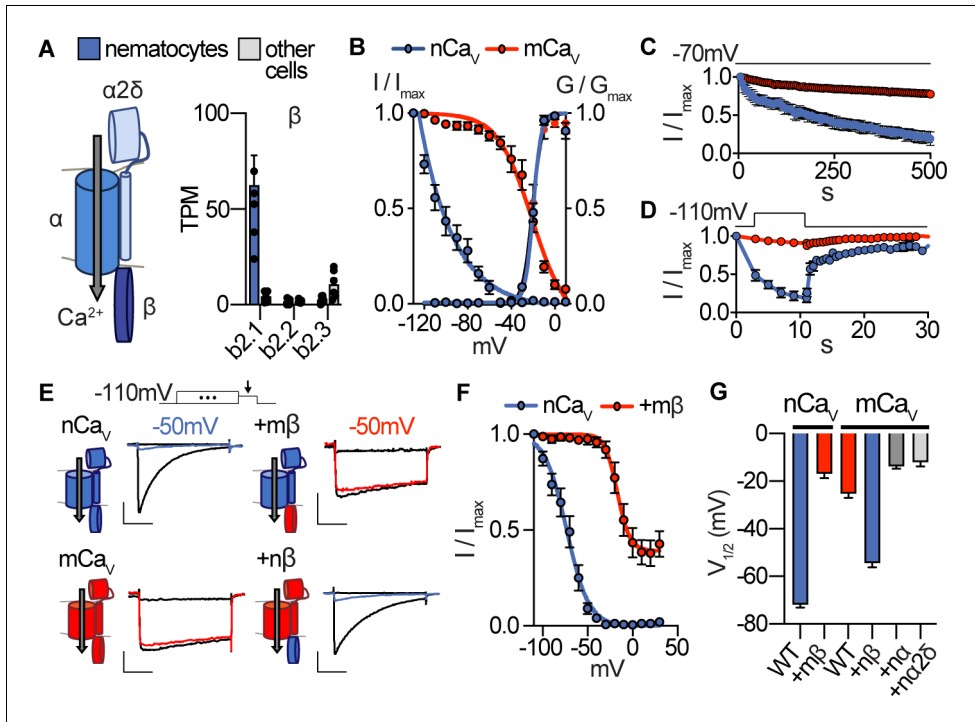

**Figure 2.** Nematostella Ca$_V$ exhibits unique voltage-dependent properties conferred by its β subunit. (**A**) Ca$_V$ channel complex with α, β, and α2δ subunits. mRNA expression (transcripts per million, TPM) of voltage-gated calcium (Ca$_V$) channel β subunits in nematocyte-enriched cells (blue) and non-enriched cells (grey). n = 6, p<0.0001 for *cacnb2.1* in nematocytes versus other cells, two-way ANOVA with post-hoc Bonferroni test. (**B**) Heterologously-expressed nCav channels (*Nematostella cacna1a, cacnb2, cacna2d1*) inactivated at very negative voltages (estimated V$_{i1/2}$ = -101.5 ± 1.6mV, n = 5) versus mammalian orthologues (mCa$_V$, V$_{i1/2}$ = -20.9 ± 3.4mV, n = 10). Apparent activation thresholds were the same: nCa$_V$ V$_{a1/2}$ = -9.8 ± 0.3mV, n = 5, mCa$_V$ V$_{a1/2}$ = -10.4 ± 0.5mV, n = 9. Inactivation was measured in response to 1 s pre-pulses from −110 mV to 10 mV with an inter-sweep holding potential of −90 mV. (**C**) nCa$_V$ exhibited slow inactivation with −70 mV holding potential (0.2 Hz stimulation, 5 s inter-pulse interval) that was best fit by two exponential functions with time constants of 10.0 and 369.5 s. n = 6, multiple row two-tailed student's t-test with significance of p<0.05 by 15 s and p<0.0001 by 500 s. (**D**) nCav inactivated at −40 mV and quickly recovered at negative holding potentials. n = 7 for nCa$_V$, n = 6 for mCa$_V$. (**E**) Voltage-gated currents recorded from nCa$_V$ or mCa$_V$ following a −110 mV pre-pulse, −50 mV pre-pulse (colored), and 20 mV pre-pulse. Ca$_V$ β subunits were substituted as indicated (mammalian β in red and *Nematostella* β in blue). Scale bars = 100 pA, 25 ms. (**F**) Mammalian β shifts nCa$_V$ voltage-dependent inactivation to positive voltages. nCa$_V$ V$_{i1/2}$ = -73.2 ± 1.2mV, n = 6. nCaV + mβ=−16.9 ± 1.9 mV, n = 6. (**G**) Half maximal inactivation voltage (V$_{i1/2}$) for Ca$_V$ chimeras. p<0.0001 for nCaV versus nCa$_V$ + mβ, mCa$_V$ versus mCa$_V$ + nβ, one-way ANOVA with post-hoc Tukey test. Inactivation was measured in response to pre-pulses from −100 mV to 10 mV with an inter-sweep holding potential of −110 mV to reduce slow inactivation. Data represented as mean ±sem.

The online version of this article includes the following source data and figure supplement(s) for figure 2:

**Source data 1.** Properties of nCa$_V$.
**Figure supplement 1.** *Nematostella* Ca$_V$ properties.

inactivation, demonstrating that channels could be reset for subsequent activation following brief exposure to negative voltage (*Figure 2D*). These distinctive features closely match the unique properties of native I$_{CaV}$, suggesting nCa$_V$ forms the predominant Ca$_V$ channel in nematocytes.

To determine the molecular basis for nCa$_V$ inactivation, we analyzed chimeric Ca$_V$ complexes containing specific α, β, and α2δ1 subunits from nCa$_V$ or mCa$_V$ orthologues. Using a holding potential of −110 mV to compare voltage-dependent inactivation, we found that only transfer of the β subunit significantly affected voltage-dependent inactivation, while α or α2δ1 subunits produced minimal effects on voltage-dependent activation, inactivation, or kinetics (*Figure 2E*, *Figure 2—figure supplement 1C–E*). Indeed, other β subunits can induce significant hyperpolarized shifts in inactivation of HVA Ca$_V$ channels (*Yasuda et al., 2004*). In this case, the mCa$_V$ β subunit drastically shifted nCa$_V$

inactivation by ~56 mV in the positive direction, prevented complete inactivation, and produced slower fast inactivation (*Figure 2E–G*, *Figure 2—figure supplement 1E*). Furthermore, nCa$_V$ β was sufficient to confer greatly enhanced voltage-dependent inactivation to mCa$_V$ (*Figure 2E,G*). From these results, we conclude that nCa$_V$ β, the most enriched Ca$_V$ subunit in nematocytes, confers nCa$_V$'s uniquely-sensitive voltage-dependent inactivation.

## Nematocyte excitability

Because electrical stimulation has been implicated in nematocyte discharge and some nematocytes can produce action potentials (*Anderson and Mckay, 1987*; *McKay and Anderson, 1988*; *Anderson and Bouchard, 2009*), we used current-clamp to record the electrical responses of nematocytes to depolarizing stimuli. Under our conditions, nematocytes had a resting potential of −64.8 ± 8.9 mV and did not produce a voltage spike when injected with current from rest (*Figure 3A*). We further considered that the strong voltage-dependent inactivation of I$_{CaV}$ could prevent excitability. Consistent with this idea, when nematocytes were first hyperpolarized to −90 mV and subsequently stimulated by current injection, we observed a singular long voltage spike (*Figure 3B,C*). In contrast, tentacle neurons produced multiple narrow spikes when injected with equivalent current amplitudes from a similar resting voltage, consistent with other neural systems (*Figure 3A–C*). Differences in spike width and frequency appear suited to mediate distinctive cellular functions: dynamic information processing in neurons and a single robust discharge event in nematocytes. Furthermore, these results indicate strong voltage-dependent inactivation prevents nematocyte activation from rest.

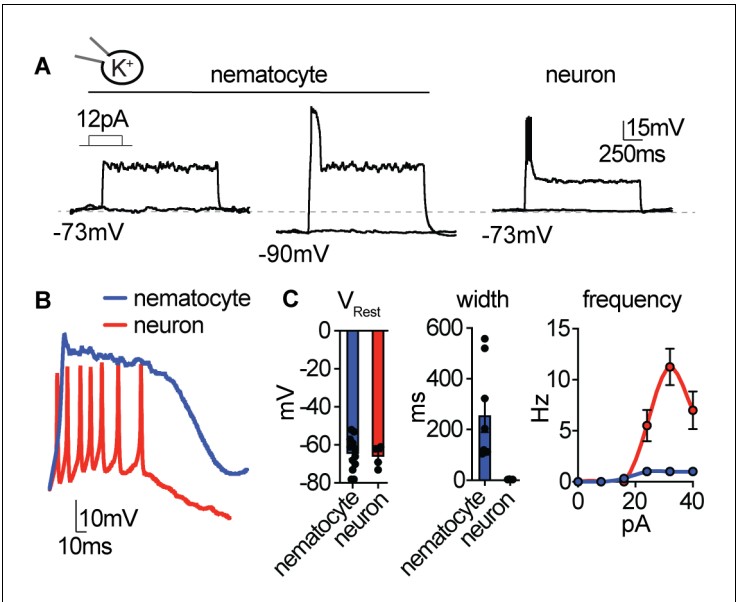

**Figure 3.** Nematocyte excitability requires hyperpolarized voltages. (**A**) Depolarizing current injection only elicited spikes from nematocytes first hyperpolarized to relieve inactivation. Nematocyte spike amplitude: 0 mV at rest, 41.5 ± 2.2 mV from ~−90 mV. n = 8, p<0.0001 two-tailed paired student's t-test. In contrast, tentacle neurons spiked from rest (31.5 ± 1.7 mV, n = 4). (**B**) Current injection elicited long singular spikes from nematocytes and numerous narrow spikes from neurons. (**C**) Nematocytes and neurons had similar resting membrane potentials but distinct spike width. n = 8 nematocytes, n=4 neurons, p<0.01, two-tailed student's t-test. Nematocytes produced only one spike, regardless of injection amplitude (n = 8), whereas neurons produced varying spike frequency depending on injection amplitude (n = 4). p<0.0001 two-way ANOVA with post-hoc Bonferroni test. Data represented as mean ±sem.

The online version of this article includes the following source data and figure supplement(s) for figure 3:

**Source data 1.** Nematocyte and neuron excitability and K$^+$currents.

**Figure supplement 1.** Nematocyte K$^+$ current properties.

$K^+$ channels contribute to resting membrane voltage (estimated reversal potential for nematocyte $K^+$ is $\sim-100$ mV) and often modulate repolarization following voltage spikes. Thus, we compared $K^+$ currents in nematocytes and tentacle neurons to understand how spike width might be differentially regulated. Nematocytes exhibited transient outward $K^+$ currents that quickly inactivated, while neurons had large sustained $K^+$ currents, perhaps important for repolarization and repetitive spiking (*Figure 3—figure supplement 1A–C*). The transient component of the nematocyte $K^+$ current was highly sensitive to voltage-dependent inactivation, similar to $I_{CaV}$ (*Figure 3—figure supplement 1D*). This $K^+$ current was abolished by the rapid intracellular $Ca^{2+}$ chelator BAPTA or by removing external $Ca^{2+}$, similar to the effect of the $K^+$ channel blocker $TEA^+$ (*Figure 3—figure supplement 1E*). Consistent with this observation, nematocyte-enriched cells expressed numerous $Ca^{2+}$-activated $K^+$ channels (*Figure 3—figure supplement 1F*). Furthermore, using intracellular $Cs^+$ to block $K^+$ currents resulted in prolonged voltage spikes and greatly increased resting membrane voltage, substantiating a role for $K^+$ channels in modulating membrane voltage (*Figure 3—figure supplement 1G*). We propose that these distinct $K^+$ channel properties could contribute to the singular wide spikes of nematocytes versus the numerous narrow spikes of neurons.

## Nematocyte sensory transduction

If nematocytes are basally inhibited due to the unique voltage-dependent inactivation of $I_{CaV}$, how do they respond to sensory signals to elicit discharge? Nematocyte discharge requires simultaneous detection of chemo- and mechanosensory cues (*Pantin, 1942b*; *Watson and Hessinger, 1992*), even though mechanical stimulation of the nematocyte's cilium (cnidocil) within intact tentacles can by itself induce cellular depolarization (*Brinkmann et al., 1996*; *Anderson and Bouchard, 2009*). Indeed, we found the deflection of isolated nematocyte cnidocils elicited a mechanically gated inward current with rapid activation and inactivation kinetics. This current was abolished by gadolinium ($Gd^{3+}$), which blocks mechanoreceptor and other cation channels, and was not observed in neurons (*Figure 4A–C*). Furthermore, nematocyte-enriched cells differentially expressed transcripts encoding NompC (no mechanoreceptor potential C, *Figure 4—figure supplement 1A*), a widely conserved mechanoreceptor previously found to localize to the cnidocil of nematocytes from *Hydra* (*Schüler et al., 2015*). Heterologous expression of *Nematostella* NompC (nNompC) resulted in a mechanically-gated current with similar properties to native nematocytes, including rapid kinetics and $Gd^{3+}$ sensitivity (*Figure 4A–C*). Comparison with the *Drosophila* orthologue (dNompC) demonstrated nNompC had similar rapid kinetics, $Gd^{3+}$ sensitivity, pressure-response relationships, and nonselective cation conductance, all consistent with the conservation of protein regions important for mechanosensitivity and ion selectivity (*Jin et al., 2017*; *Figure 4A–C*, *Figure 4—figure supplement 1B–D*). Thus, we conclude nematocytes are intrinsically mechanosensitive and suggest nNompC contributes to nematocyte mechanosensitivity. Importantly, this mechanically-evoked current is of sufficient amplitude to evoke a spike from very negative membrane voltages, but not from resting voltage at which $I_{CaV}$ is inactivated.

Because nematocyte discharge is mediated by combined chemical and mechanical cues (*Pantin, 1942b*; *Watson and Hessinger, 1992*), we wondered if chemosensory signals could modulate nematocyte membrane voltage to allow for $I_{CaV}$ activation and cellular responses. While prey-derived chemicals (<3 kDa extract from brine shrimp) evoked robust behavioral responses, similar treatments did not elicit electrical responses from isolated nematocytes (*Figure 4D*, *Figure 4—figure supplement 1E*). Considering in vivo cellular and discharge activity requires the presence of prey-derived chemicals with simultaneous mechanical stimulation, chemoreception may occur indirectly through functionally coupled cells (*Price and Anderson, 2006*). Previous studies suggest the presence of synaptic connections between nematocytes and other unknown cell types (*Westfall et al., 1998*; *Oliver et al., 2008*). To test this possibility, we screened isolated nematocytes for responses to well-conserved neurotransmitters and found only acetylcholine (ACh) elicited a significant response (*Figure 4D,E*). ACh-evoked outward currents were abolished by nicotinic acetylcholine receptor (nAChR) antagonists and recapitulated by nicotine (*Figure 4F*, *Figure 4—figure supplement 1F*). The $K^+$ channel blocker $TEA^+$ and the intracellular $Ca^{2+}$ chelator BAPTA inhibited responses, suggesting ACh elicits $K^+$ channel activity downstream of increased intracellular $Ca^{2+}$. While the G-protein signaling inhibitor GDPβS did not affect outward currents, blockade of $K^+$ currents with intracellular $Cs^+$ revealed an ACh-elicited inward current that was enhanced by increased extracellular $Ca^{2+}$ and blocked by the nAChR antagonist mecamylamine (*Figure 4F*, *Figure 4—*

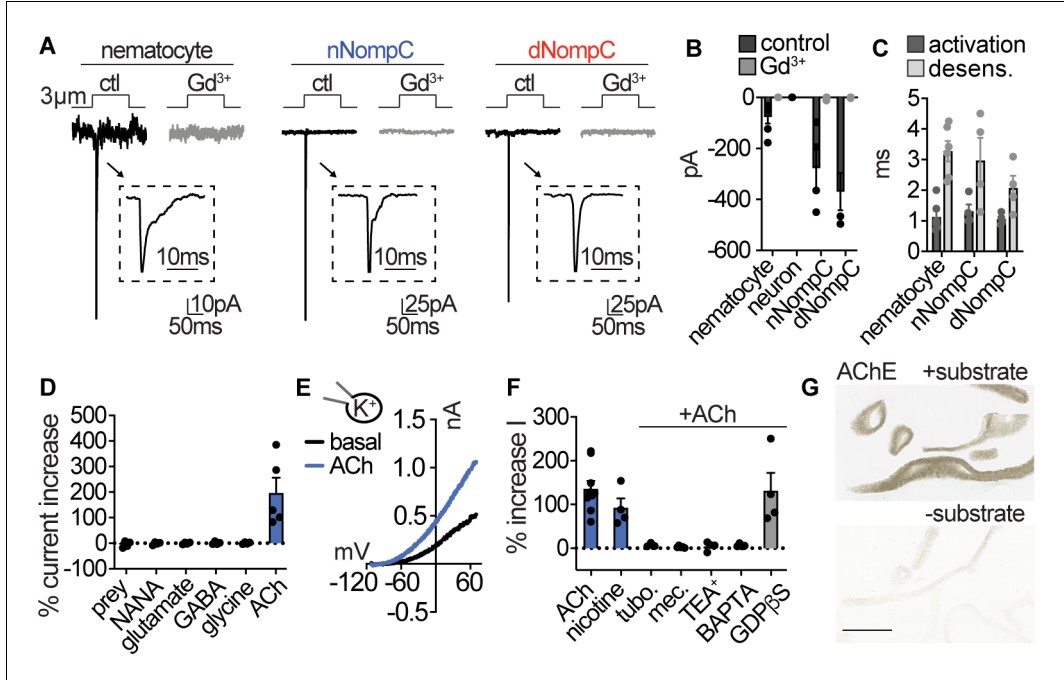

**Figure 4.** Nematocytes are intrinsically mechanosensitive and indirectly chemosensitive. (**A**) Mechanical stimulation of nematocytes evoked $Gd^{3+}$-sensitive, transient inward currents with similar properties to heterologously expressed *Nematostella* (n) and *Drosophila* (d) NompC channels. Stimulation thresholds (pipette displacement): nematocyte = 1.7 ± 0.3 µm (n = 6), nNompC = 4.2 ± 0.6 µm (n = 4), dNompC = 4.2 ± 0.5 µm (n = 4). Untransfected cells did not respond to similar stimuli (n = 6). (**B**) Mechanically evoked currents from nematocytes (n = 6), nNompC (n = 4), and dNompC (n = 4) were blocked by $Gd^{3+}$, while tentacle neurons lacked mechanically evoked currents (n = 8). p<0.01 two-tailed student's t-test. (**C**) Mechanically evoked current activation and desensitization kinetics were similar in nematocytes (n = 6), nNompC (n = 4), and dNompC (n = 4). (**D**) Chemosensory stimuli did not directly affect nematocytes but the neurotransmitter acetylcholine (ACh, n = 5) elicited a large outward current. n = 4 for prey extract, NANA, Glutamate, GABA, Glycine, p<0.001 for ACh versus other conditions, one-way ANOVA with post-hoc Tukey test. (**E**) Representative current-voltage relationship of ACh-elicited response in nematocytes. (**F**) ACh-evoked currents (n = 9) were blocked by nicotinic ACh receptor antagonists (tubocurarine = 4, mecamylamine = 4) and a similar current was elicited by nicotine (n = 4). ACh-evoked outward currents were inhibited by a $K^+$ channel blocker (TEA$^+$, n = 4) and an intracellular $Ca^{2+}$ chelator (BAPTA, n = 4), but not the G-protein signaling blocker GDPβS (n = 4). p<0.001 for vehicle versus antagonists, one-way ANOVA with post-hoc Tukey test. (**G**) Acetylcholinesterase staining in tentacles with and without substrate solution (representative of n = 3 animals). Scale bar = 200 µm. Data represented as mean ±sem.

The online version of this article includes the following source data and figure supplement(s) for figure 4:

**Source data 1.** Mechano- and chemosensory properties.

**Figure supplement 1.** Sensory transduction properties.

*figure supplement 1G,H*). These results suggest ACh evokes a $Ca^{2+}$-permeable nAChR-like signaling pathway to engage $Ca^{2+}$-activated $K^+$ channels, consistent with the absence of muscarinic ACh receptors in *Nematostella* (*Faltine-Gonzalez and Layden, 2019*). In agreement with this observation, nematocyte-enriched cells expressed numerous nAChR-like transcripts which had well-conserved domains involved in $Ca^{2+}$ permeability (*Fucile, 2004*; *Figure 4—figure supplement 1I,J*). Finally, we found robust acetylcholinesterase activity in tentacles, further suggesting a role for ACh signaling in nematocyte function (*Figure 4G*). These results demonstrate that nematocytes use cholinergic signaling to regulate $K^+$ currents, similar to how efferent cholinergic innervation of vertebrate hair cells modulates nAChR-$K^+$ channel signaling to inhibit auditory responses (*Elgoyhen and Katz, 2012*).

To identify the origin of cellular connections to nematocytes, we used serial electron microscopy reconstruction to visualize nematocytes and neighboring cells. We analyzed similar tentacle tissues from which we carried out physiological experiments and readily observed neurons and nematocytes

in close proximity (*Figure 5—figure supplement 1A,B*). In resulting micrographs, nematocytes were clearly identified by their distinct nematocyst and cnidocil (*Figure 5—figure supplement 1C*). Interestingly, each nematocyte exhibited a long process, of presently unknown function, that extended into the ectoderm (*Figure 5—figure supplement 1D*). We also observed numerous spirocytes, indicated by the presence of a large intracellular thread-like structure (*Figure 5A*, *Figure 5—figure supplement 1C*). Putative sensory neurons were identified based on their synaptic contacts and extracellular projections (*Figure 5A*, *Figure 5—figure supplement 1E,F*). Importantly, dense core

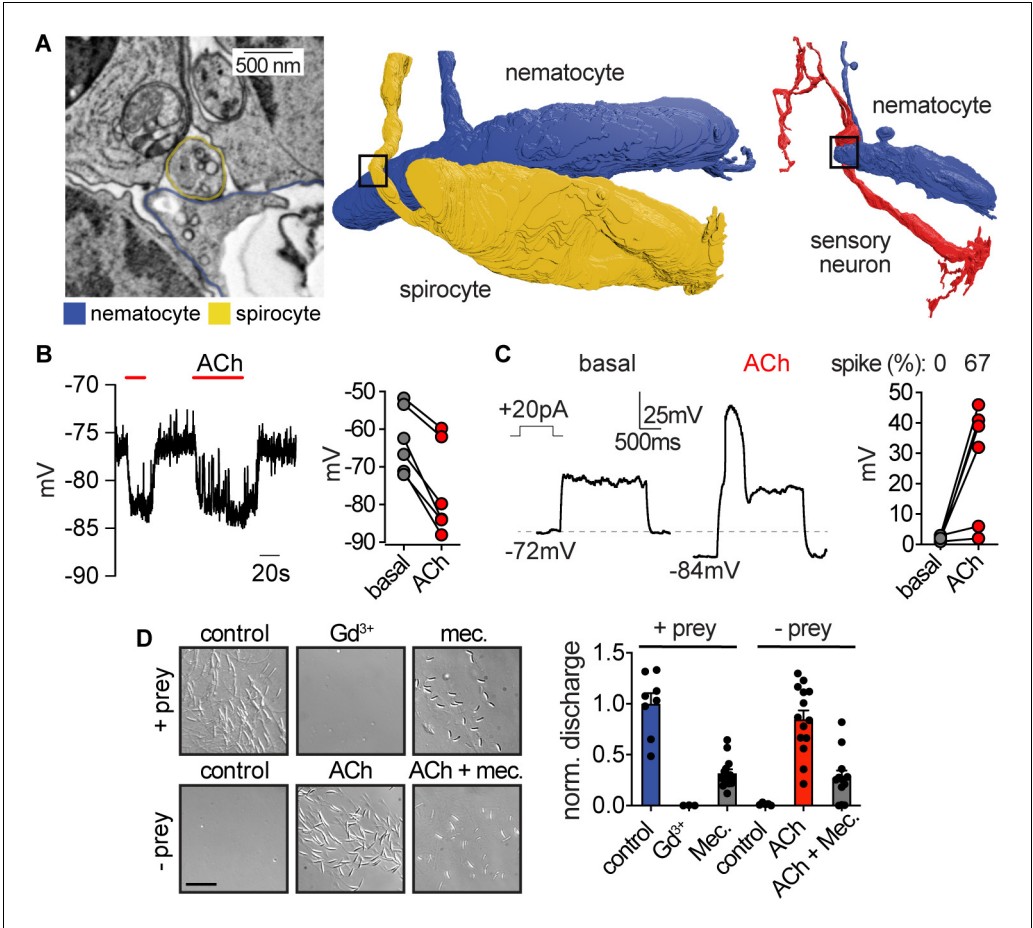

**Figure 5.** Nematocyte voltage-dependence facilitates signal integration required for stinging. (**A**) *Left*: Electron micrograph demonstrating dense-core vesicles in the vicinity of an electron-dense zone localized at the junction between a spirocyte and nematocyte. *Middle*: 3D reconstruction of the nematocyte and spirocyte shown in the left panel. The box indicates the position of the synapse. *Right*: 3D reconstruction of a different nematocyte making a synapse with a putative sensory neuron. The box indicates the position of the synapse shown in *Figure 5—figure supplement 2C and E*. (**B**) ACh induced hyperpolarization of nematocytes. n = 6, p<0.01 paired two-tailed student's t-test. (**C**) Depolarizing current injection did not induce active properties from rest, but did elicit a voltage spike from most nematocytes hyperpolarized by ACh. n = 6, p<0.01 paired two-tailed student's t-test. (**D**) Chemo-tactile-induced nematocyte discharge (touch + prey extract, n = 8) was inhibited by the mechanoreceptor current blocker $Gd^{3+}$ (n = 7) and nAChR antagonist mecamylamine (n = 14). In the absence of chemical stimulation (touch - prey extract, n = 5), touch + ACh (n = 14) was sufficient to induce discharge, which was inhibited by mecamylamine (n = 12). p<0.0001 for controls versus respective treatments, one-way ANOVA with post-hoc Bonferroni test. Data represented as mean ±sem.

The online version of this article includes the following source data and figure supplement(s) for figure 5:

**Source data 1.** Combinatorial sensory cues modulate discharge.
**Figure supplement 1.** Nematocyte morphology.
**Figure supplement 2.** Nematocyte synaptic connections.
**Figure supplement 3.** Nematocyte signaling pathways.

vesicles were localized to electron-dense regions at the junction between each nematocyte and one other cell type, either sensory neurons or spirocytes (*Figure 5A*, *Figure 5—figure supplement 2A–E*). Thus, nematocytes receive synaptic input from both neurons and spirocytes and likely serve as a site for integrating multiple signals (*Figure 5—figure supplement 2F,G*). This observation is consistent with the ability of cnidarians to simultaneously discharge multiple cnidocyte types to most efficiently capture prey (*Pantin, 1942a*).

### Voltage-dependence mediates signal integration

How do distinct mechanosensory and chemosensory signals converge to elicit discharge? In agreement with our observation that nAChR activation increases $K^+$ channel activity, ACh hyperpolarized nematocytes to negative voltages from which they were capable of producing robust voltage spikes (*Figure 5B,C*). A select number of cells with a more positive resting voltage still failed to produce spikes, and therefore additional regulation could exist through the modulation of resting membrane voltage (*Figure 5—figure supplement 3A,B*). These results suggest that the voltage-dependence for $I_{CaV}$ prevents basal activation to depolarizing signals, such as mechanical stimulation, but activation of nAChR hyperpolarizes the cell to relieve $I_{CaV}$ inactivation, thereby amplifying depolarizing signals to mediate cellular responses.

Consistent with a requirement for both mechano- and chemosensory input, we found the mechanoreceptor current blocker $Gd^{3+}$ inhibited chemo-tactile stimulation of discharge. Additionally, the nAChR antagonist mecamylamine greatly reduced chemo-tactile-induced discharge (*Figure 5D*). Washout of both treatments recovered the ability of nematocytes to discharge (*Figure 5—figure supplement 3C*). Moreover, the requirement for prey-derived chemicals was completely recapitulated by ACh (*Figure 5D*). These results are consistent with a role for ACh signaling downstream of chemosensory stimulation. Thus, we propose that the unique $I_{CaV}$ voltage-dependent inactivation provides a mechanism by which nematocytes filter extraneous depolarizing mechanical signals, but can integrate chemosensory-induced hyperpolarization together with a depolarizing stimulus to elicit robust signal amplification and discharge responses (*Figure 6*).

## Discussion

Here, we demonstrate that nematocytes use the specialized properties of $nCa_V$ to filter salient chemo-tactile signals from environmental noise. The involvement of $Ca^{2+}$ signaling in this process is consistent with a well-established role for $Ca^{2+}$ influx in mediating discharge across multiple cnidarian species (*McKay and Anderson, 1988*; *Santoro and Salleo, 1991*; *Watson and Hessinger, 1994a*). However, the exact mechanism by which $Ca^{2+}$ influx mediates discharge is unclear. It has been proposed that $Ca^{2+}$ influx alters the permeability of the nematocyst capsule and/or initiates the rapid dissociation of ions within the cyst to induce osmotic changes within the organelle (*Lubbock and Amos, 1981*; *Lubbock et al., 1981*; *Weber, 1990*; *Tardent, 1995*). Our transcriptomic analyses demonstrate that nematocytes express multiple $Ca^{2+}$ handling proteins (*Figure 5—*

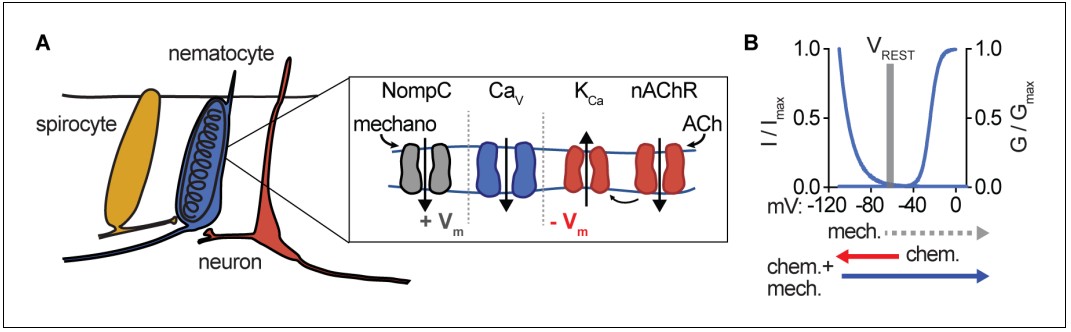

**Figure 6.** Nematocyte $Ca_V$ properties filter salient chemo-tactile signals. (**A**) Model for nematocyte signal integration. (**B**) Nematocyte $Ca_V$ is inactivated at rest and thus does not amplify extraneous NompC-mediated mechanical signals. Chemosensory stimuli hyperpolarize nematocytes through ACh signaling to relieve inactivation of $Ca_V$ channels, which can then amplify mechanical stimuli to engage discharge.

figure supplement 3D–F), which could control $Ca^{2+}$ signaling domains in response to spatially restricted sensory transduction cascades. This organization would be consistent with the necessity for $I_{CaV}$-mediated amplification of receptor-mediated nonselective cation conductances. Indeed, $Ca_V$ currents could mediate an increase of >500 µM $Ca^{2+}$ considering a uniform distribution across the small cytoplasmic volume (5%) not occupied by the nematocyst. Future studies will provide insight into the coupling between sensory transduction and organellar physiology.

Our results suggest one mechanism by which nematocytes integrate combined mechanical and chemical cues to filter salient environmental information and appropriately engage nematocyte discharge. However, cnidarians occupy distinct ecological niches and may have evolved different biophysical features to account for increased turbulence, specific behaviors, or particular prey and predatory targets. Numerous sea anemones occupy turbulent tidal pools, whereas others, like *Nematostella vectensis,* live in calmer regions. Similarly, cnidarians can undergo developmental transitions between immobile and free-floating medusa phases while maintaining the use of nematocytes for prey capture (*Martin and Archer, 1997*). Although forces generated from swimming prey are likely negligible in comparison with physical contact of the cnidocil, strong tidal waves may be sufficient to elicit mechanoreceptive responses which could interfere with pertinent chemo-tactile sensation and subsequent stinging responses. These ecological differences might require distinct filtering mechanisms for distinguishing salient prey or predator signals. For example, anatomical organization and pharmacological dependence for cellular and discharge activity in anthozoan nematocytes differs from hydrozoans (*Anderson and Mckay, 1987*; *Kass-Simon and Scappaticci, Jr., 2002*; *Oliver et al., 2008*). Indeed, nematocysts vary extensively in morphology, differing in the length of the extruded thread, the presence of spines, and the composition of toxins (*Kass-Simon and Scappaticci, Jr., 2002*), likely reflecting the diversity of organismal needs.

The modularity provided by synaptic connections could increase the diversity of signals which regulate nematocyte discharge. For instance, distinct chemoreceptor cells could form synaptic connections with specific nematocyte populations to mediate discrete behavioral responses. In addition to chemical mixtures, such as prey extract, numerous amino acids, lipids, and N-acetylated sugars can individually modulate nematocyte discharge (*Watson and Hessinger, 1992*). While these are broadly-distributed molecules, it is possible distinct prey- or predator-derived compounds regulate specific chemosensory cells to engage predatory or defensive nematocytes, respectively (*Brace, 1990*). Indeed, we observed that both spirocytes and neurons make synaptic contacts with nematocytes, thus either could release ACh in response to specific stimuli. Future identification and characterization of chemosensory cells and the ecologically relevant compounds which activate them will provide insight regarding chemical coding and mechanisms of synaptic signaling. Discharge can also be regulated by organismal nutritional state (*Sandberg et al., 1971*), suggesting nematocytes could receive input from digestive cells or hormones. Additionally, various cnidocyte types are found across the cnidarian phylum and regulated by stimuli relevant to their behavioral purpose. For example, the freshwater cnidarian, *Hydra vulgaris*, uses a specific cnidocyte to grasp surfaces for phototaxis, suggesting that these cells could be regulated downstream of a photoreceptor (*Plachetzki et al., 2012*). Within anthozoans, nematocytes and spirocytes may use similar or distinct mechanisms to control discharge. Functional comparisons will reveal whether specific proteins, domains, or signaling mechanisms are conserved or give rise to the evolutionary novelties across these incredibly specialized cell types (*Babonis and Martindale, 2014*).

The ability to distinguish behaviorally-relevant stimuli, such as prey, from background noise is especially critical because nematocytes are single-use cells that must be replaced following discharge. Multiple species have taken advantage of these specialized conditions by adapting to evade and exploit nematocyte discharge for their own defensive purposes. For example, clownfish can live among the tentacles of sea anemones without harmful effects, although the exact mechanism by which this occurs is unclear (*Lubbock, 1980*). Certain species of nudibranchs and ctenophores acquire undischarged nematocysts from prey and store them for later defense, indicating that these organisms are able to initially prevent discharge responses (*Greenwood, 2009*). Understanding such regulation could reveal additional mechanisms by which cells process diverse stimuli and provide insight into the evolution of these interspecies relationships.

# Materials and methods

## Key resources table

| Reagent type (species) or resource | Designation | Source or reference | Identifiers | Additional information |
|---|---|---|---|---|
| Strain, strain background (*Nematostella vectensis,* adult, male and female) | *Nematostella vectensis* | Woods Hole Marine Biological Laboratory | | |
| Strain, strain background (*Nematostella vectensis,* stable transgenic line, adult, male and female) | NvElav1::mOrange | (**Nakanishi et al., 2012**) | | |
| Cell line (*Homo-sapiens*) | HEK293T | ATCC | | CRL-3216 |
| Antibody | Anti- DsRed (Rabbit Polyclonal) | TaKaRa | Cat#632496 | 1:500 |
| Recombinant DNA reagent | nCa$_V$: cacna1a (plasmid) | Plasmid from our lab (see Materials and methods) | | Nematostella nCa$_V$ α |
| Recombinant DNA reagent | nCa$_V$: cacna2d1 (plasmid) | Plasmid from our lab (see Materials and methods) | | Nematostella nCa$_V$ α2δ |
| Recombinant DNA reagent | nCa$_V$: cacnb2 (plasmid) | Plasmid from our lab (see Materials and methods) | | Nematostella nCa$_V$ β |
| Recombinant DNA reagent | nNompC (plasmid) | Plasmid from our lab (see Materials and methods) | | Nematostella nNompC |
| Recombinant DNA reagent | Rat *cacna2d1* | (**Lin et al., 2004**) | Addgene Plasmid #26575 | Rat mCa$_V$ α2δ |
| Recombinant DNA reagent | Mouse *cacna1a* | (**Richards et al., 2007**) | Addgene Plasmid #26578 | Mouse mCa$_V$ α |
| Recombinant DNA reagent | Rat *cacnb2a* | (**Wyatt et al., 1998**) | Addgene Plasmid #107424 | Rat mCa$_V$ β |
| Recombinant DNA reagent | dNompC-GFP | YN Jan (UCSF) | N/A | *Drosophila* nNompC |
| Sequence-based reagent | RACE Primer | Primer generated in our lab (see Materials and methods) | PCR primer | GATTACGCCAAGCTTTATGCG TCCAATCGTACTTGTCGGC |
| Sequence-based reagent | RACE Primer | Primer generated in our lab (see Materials and methods) | PCR primer | GATTACGCCAAGCTTGCCGA CAAGTACGATTGGACGCATA |
| Sequence-based reagent | Cacnb primer F | Primer generated in our lab (see Materials and methods) | PCR primer | CAGAGCCAGGCCTGAGCGAG |
| Sequence-based reagent | Cacnb primer R | Primer generated in our lab (see Materials and methods) | PCR primer | GCCCCGTTAAAAGTCGAGAG |
| Commercial assay or kit | SMARTer RACE 5'/3' Kit | TaKaRa | Cat# 634858 | |
| Chemical compound, drug | N-Acetylneuraminic acid (NANA) | Sigma | Cat#857459 | |
| Chemical compound, drug | Glycine | Sigma | Cat#410225 | |

*Continued on next page*

*Continued*

| Reagent type (species) or resource | Designation | Source or reference | Identifiers | Additional information |
|---|---|---|---|---|
| Chemical compound, drug | Acetylcholine chloride | Sigma | Cat#A6625 | |
| Chemical compound, drug | GdCl$_3$ | Sigma | Cat#7532 | |
| Chemical compound, drug | TEA | Sigma | Cat#86614 | |
| Chemical compound, drug | GDPβS | Sigma | Cat#G7637 | |
| Chemical compound, drug | CdCl$_2$ | Sigma | Cat#202908 | |
| Chemical compound, drug | Mecamylamine | Tocris | Cat#2843 | |
| Chemical compound, drug | Glutamate | Tocris | Cat#0218 | |
| Chemical compound, drug | γ-Aminobutyric acid (GABA) | Tocris | Cat#0344 | |
| Chemical compound, drug | Nicotine ditartrate | Tocris | Cat#3546 | |
| Chemical compound, drug | Tubocurarine | Tocris | Cat#2820 | |
| Chemical compound, drug | BAPTA | Molecular Probes | Cat#B-1204 | |
| Software, algorithm | Cutadapt | (*Martin, 2011*) | | https://github.com/marcelm/cutadapt/ |
| Software, algorithm | Trinity | (*Grabherr et al., 2011*) | | https://github.com/trinityrnaseq |
| Software, algorithm | InterProScan | (*Jones et al., 2014*) | | https://github.com/ebi-pf-team/interproscan/ |
| Software, algorithm | HMMER | (*Eddy, 2009*) | | http://hmmer.org/ |
| Software, algorithm | Pfam | (*El-Gebali et al., 2019*) | | https://pfam.xfam.org/ |
| Software, algorithm | DIAMOND | (*Buchfink et al., 2015*) | | https://github.com/bbuchfink/diamond |
| Software, algorithm | Clustal Omega | (*Sievers et al., 2011*) | | https://www.ebi.ac.uk/Tools/msa/clustalo/ |
| Software, algorithm | Fiji: Linear Stack Alignment with SIFT | (*Schindelin et al., 2012*) | | https://fiji.sc/ |
| Software, algorithm | Matlab custom script | Matlab code provided in source data for *Figure 1* | | See supplemental table for script |
| Software, algorithm | VAST | (*Berger et al., 2018*) | | https://software.rc.fas.harvard.edu/lichtman/vast/ |
| Software, algorithm | 3D Studio Max 2019 | (*Autodesk, 2019*) | | https://www.autodesk.com |

## Animals and cells

Starlet sea anemones (*Nematostella vectensis*) were provided by the Marine Biological Laboratory (Woods Hole, Massachusetts), Nv-Elav1::mOrange transgenic animals were a gift from F. Rentzsch. We used adult animals of both sexes, which were fed freshly-hatched brine shrimp (*Artemia*) twice a week and kept on a 12 hr light/dark cycle in 1/3 natural sea water (NSW). Nematocytes and neurons were isolated from tentacle tissue, which was harvested by anesthetizing animals in high-magnesium solution containing (mM): 140 NaCl, 3.3 Glucose, 3.3 KCl, 3.3 HEPES, 40 MgCl$_2$. Cells were isolated from tentacles immediately prior to electrophysiology experiments by treatment with 0.05% Trypsin at 32°C for 30 min and mechanical dissociation in divalent free recording solution (mM): 140 NaCl,

3.3 Glucose, 3.3 KCl, 3.3 HEPES, pH 7.6. Basitrichous isorhiza nematocytes were isolated from tentacles and identified by a thick-walled capsule containing a barbed thread, with a characteristic high refractive index, oblong shape and presence of a cnidocil. Spirocytes were identified by a thin-walled capsule containing a thin, unarmed thread, used for ensnaring prey. Neurons were identified by mOrange expression.

HEK293T cells (ATCC, Cat# CRL-3216, RRID:CVCL_0063, authenticated and validated as negative for mycoplasma by vendor) were grown in DMEM, 10% fetal calf serum, and 1% penicillin/streptomycin at 37°C, 5% $CO_2$. Cells were transfected using lipofectamine 2000 (Invitrogen/Life Technologies) according to the manufacturer's protocol. 1 μg of *Nematostella cacna1a*, *cacnb2.1*, *cacna2d1*. *M. musculus* (mouse) *cacna1a*, *R. norvegicus* (rat) *cacnb2a*, or rat *cacna2d1* was coexpressed with 0.5 μg GFP. Mechanosensitive proteins were assayed using HEK293T cells transfected with 1 μg of either *Drosophila* NompC or *Nematostella* NompC. To enhance channel expression, cells were transfected for 6–8 hr, plated on coverslips, and then incubated at 28°C for 2–6 days before experiments. *Drosophila* NompC-GFP was a gift from YN Jan. Rat *cacna2d1* (RRID:Addgene_26575) and *cacna1a* (RRID:Addgene_26578) were gifts from D. Lipscombe and *cacnb2a* (RRID:Addgene_107424) was a gift from A. Dolphin.

## Molecular biology

RNA was prepared from tentacles of adult *Nematostella* using published methods (*Stefanik et al., 2013*). Briefly, 50 mg of tentacle tissue were homogenized and RNA was extracted using TRIzol. RNA was isolated and DNase treated (Zymo Research), then used for cDNA library synthesis (NEB E6560). Full-length sequence for a *Nematostella* calcium channel beta subunit was obtained with a RACE strategy using specific primers (GATTACGCCAAGCTTTATGCGTCCAATCGTACTTGTCGGC and GATTACGCCAAGCTTGCCGACAAGTACGATTGGACGCATA) on the amplified tentacle-tissue library. The final sequence was confirmed using primers corresponding to the end of the derived sequence (CAGAGCCAGGCCTGAGCGAG and GCCCCGTTAAAAGTCGAGAG) to amplify a full-length cDNA from tentacle mRNA, which was sequenced to confirm identity. $nCa_v$ subunits: cacna1a, cacna2d1, cacnb2, and nNompC-GFP were synthesized by Genscript (Piscataway, NJ). Sequence alignments were carried out using Clustal Omega.

## Transcriptomics

Tentacle tissue was ground to a fine powder in the presence of liquid nitrogen in lysis buffer (50 mM Tris-HCl pH 7.5, 250 mM KCl, 35 mM $MgCl_2$, 25 mM EGTA-KOH pH 8, 5 mM DTT, murine RNase inhibitor (NEB), 1% (w/v) NP-40, 5% (w/v) sucrose, 100 μg $ml^{-1}$ cycloheximide (Sigma), 500 μg $ml^{-1}$ heparin (Sigma)). Lysate was incubated on ice for 5 min, triturated five times with an 18 g needle, and insoluble material was removed by centrifugation at 16,000 g for 5 min at 4°C. Polyadenylated RNA was used to make sequencing libraries and sequenced on an Illumina HiSeq 4000 (Novogene). Quality filtering and adapter trimming was performed using Cutadapt (*Martin, 2011*), and a de novo transcriptome was assembled using Trinity (*Grabherr et al., 2011*). Annotation was performed using InterProScan (*Jones et al., 2014*) with Panther member database analysis, HMMER (*Eddy, 2009*) with the Pfam (*El-Gebali et al., 2019*) database, and DIAMOND (*Buchfink et al., 2015*) with the UniProt/TrEMBL database.

Reads from sorted cnidocytes (Bioproject PRJNA391807 *Sunagar et al., 2018*), enriched for nematocytes, were quality and adapter trimmed as described above, and transcript abundance (TPM) was quantified using Kallisto (*Bray et al., 2016*) and the tentacle transcriptome. For read mapping visualization, mapping was performed with Bowtie2 (*Langmead and Salzberg, 2012*), output files were converted to indexed bam files using Samtools (*Li et al., 2009*), and visualization was performed with the Integrated Genomics Viewer (*Robinson et al., 2011*).

## Electrophysiology

Recordings were carried out at room temperature using a MultiClamp 700B amplifier (Axon Instruments) and digitized using a Digidata 1550B (Axon Instruments) interface and pClamp software (Axon Instruments). Whole-cell recording data were filtered at 1 kHz and sampled at 10 kHz. For single-channel recordings, data were filtered at 2 kHz and sampled at 20 kHz. Data were leak-subtracted online using a p/4 protocol, and membrane potentials were corrected for liquid junction

potentials. For whole-cell nematocyte and neuron recordings, borosilicate glass pipettes were polished to 8–10 MΩ. The standard *Nematostella* medium was used as the extracellular solution and contained (in mM): 140 NaCl, 3.3 glucose, 3.3 KCl, 3.3 HEPES, 0.5 CaCl$_2$, 0.5 MgCl$_2$, pH 7.6. Two intracellular solutions were used for recording. For isolating inward currents (mM): 133.3 cesium methanesulfonate, 1.33 MgCl$_2$, 3.33 EGTA, 3.33 HEPES, 10 sucrose, 10 CsEGTA, pH 7.6. For outward currents (mM) 166.67 potassium gluconate, 3.33 HEPES, 10 sucrose, 1.33 MgCl$_2$, 10 KEGTA, pH 7.6. In some experiments, BAPTA was substituted for EGTA. For whole-cell recordings in HEK293 cells, pipettes were 3–4 MΩ. The standard extracellular solution contained (in mM): 140 NaCl, 5 KCl, 10 HEPES, 2 CaCl$_2$, 1 MgCl$_2$, pH 7.4. The intracellular solution contained (mM): 140 cesium methanesulfonate, 1 MgCl$_2$, 3.33 EGTA, 3.33 HEPES, 10 sucrose, pH 7.2. In ion substitution experiments, equimolar Ba$^{2+}$ was substituted for Ca$^{2+}$. Single-channel recording extracellular solution contained (mM): 140 NaCl, 10 HEPES, 1 NaEGTA, pH 7.4. The intracellular solution used (mM): 140 CsCl, 10 HEPES, 1 CsEGTA, pH 7.4.

The following pharmacological agents were used: N-Acetylneuraminic acid (NANA, 100 µM, Sigma), glycine (100 µM, Sigma), acetylcholine (1 mM), mecamylamine (100 µM, 500 µM for behavioral experiments, Tocris), GdCl$_3$ (100 µM, Sigma), glutamate (1 mM), GABA (1 mM), nicotine (100 µM, Tocris), tubocurarine (10 µM, Tocris), TEA$^+$ (10 mM, Sigma), BAPTA (10 mM, Tocris), GDPβS (1 mM, Sigma), and Cd$^{2+}$ (500 µM, 250 µM for behavioral experiments). All were dissolved in water. Prey extract was isolated from freshly-hatched *Artemia*. *Artemia* were flash-frozen, ground with mortar and pestle, filtered with 0.22 µM pores or 3 kDa ultracentrifugal filters (Amicon UFC500324). Pharmacological effects were quantified as differences in normalized peak current from the same cell following bath application of the drug (I$_{treatment}$/I$_{control}$). Whole-cell recordings were used to assess mechanical sensation together with a piezoelectric-driven (Physik Instrumente) fire-polished glass pipette (tip diameter 1 µm). Mechanical steps in 0.5 µm increment were applied every 5 s while cells were voltage-clamped at −90 mV. Single mechanosensitive channels were studied using excised outside-out patches exposed to pressure applied via a High-Speed Pressure Clamp system (HSPC, ALA-scientific). Pressure-response relationships were established using pressure steps in 10 mmHg increments. Voltage-dependence of currents was measured from −100 mV to 100 mV in 20 mV increments while applying repetitive 60 mmHg pressure pulses.

Unless stated otherwise, voltage-gated currents were measured in response to a 200 ms voltage pulse in 10 mV increments from a −110 mV holding potential. G-V relationships were derived from I-V curves by calculating G: G = I$_{CaV}$/(V$_m$-E$_{rev}$) and fit with a Boltzman equation. Voltage-dependent inactivation was measured during −10 mV (Ca$^{2+}$ currents in native cells), 0 mV (Ca$^{2+}$ currents in heterologously expressed channels), 60 mV (K$^+$ currents in native cells) voltage pulses following a series of 1 s pre-pulses ranging from −110 mV to 60 mV. Voltage-dependent inactivation was quantified as I/I$_{max}$, with I$_{max}$ occurring at the voltage pulse following a −110 mV prepulse. In some instances, inactivation curves could not be fit with a Boltzman equation and were instead fitted with an exponential. The time course of voltage-dependent inactivation was measured by using a holding voltage of −110 mV or −70 mV and applying a 0 mV test pulse every 5 s. Recovery from inactivation was quantified by normalizing inactivated and recovered currents to those elicited from the same cell in which a 0 mV voltage pulse was applied from −110 mV. Test pulses from a holding voltage of −40 mV were used to assess inactivation, followed by pulses from −110 mV where currents quickly recovered to maximal amplitude. Repetitive stimulation using 20 ms pulses to −10 mV or 0 mV from a holding voltage of −90 mV was also used to measure inactivation in response to repetitive stimulation. Current inactivation kinetics were quantified by the portion of current remaining at the end of a 200 ms pulse (R200) or fit with a single exponential. Activation was quantified as the time from current activation until peak amplitude. 200 ms voltage ramps from −120 to 100 mV were used to measure ACh-elicited currents. Stimulus-evoked currents were normalized to basal currents measured at the same voltage of 80 mV.

Single channel currents were measured from the middle of the noise band between closed and open states or derived from all-points amplitude histograms fit with Gaussian relationships at closed and open peaks for each excised patch record. Conductance was calculated from the linear slope of I–V relationships. N(P$_O$) was calculated during pressure steps while voltage was held at −80 mV. In current clamp recordings, effects of ACh or intracellular ions on resting membrane potential was measured without current injection (I = 0). 1 s depolarizing current steps of various amplitudes were injected to measure spikes which were quantified by frequency (spikes/second) or width (duration of

spike). To test whether resting membrane potential affects the ability to generate spikes, hyperpolarizing current was injected to bring cells to negative voltages (<-90mV) or ACh was locally perfused before depolarizing current injection.

The change in $Ca^{2+}$ concentration from a nematocyte voltage spike was estimated based on the integral of $Ca^{2+}$-selective nematocyte currents elicited by a 0 mV step, the same amplitude and slightly shorter duration than a voltage spike. Nematocyte volume was estimated from serial electron microscopy reconstruction with a non-nematocyst volume of approximately 5% of the total volume of the cell. We did not consider the volume occupied by other organelles, making for a conservative estimate. Furthermore, calculations were made with extracellular recording solution containing 0.5 mM $Ca^{2+}$, which is approximately six-fold less than physiological concentrations. Thus, the large increase we calculated likely underestimates the total $Ca^{2+}$ influx.

## Immunohistochemistry
### Neural staining
Adult Nv-Elav1::mOrange *Nematostella* were paralyzed in anesthetic solution, then placed in a 4% solution of PFA overnight. Animals were cryoprotected using a gradient of increasing sucrose concentrations (10% to 50%) in PBS over two days. Cryostat sections (20 µm thick) were permeabilized with 0.2% Triton-X and 4% normal goat serum (NGS) at room temperature for 1 hr, followed by incubation with DsRed Polyclonal Antibody (Takara Bio Cat# 632496, RRID:AB_10013483) overnight in PBST (0.2%) and NGS (4%) at 4°C. Tissue was rinsed three times with PBST before secondary was applied (Goat anti-rabbit 647, Abcam in PBST + NGS) for 2 hr at room temperature. Tissue was rinsed with PBS and mounted with Vectashield containing DAPI (Novus Biologicals).

### Acetylcholinesterase staining
Tentacles were stained for the presence of acetylcholinesterase as described (*Paul et al., 2010*) using 40 µm thick cryosections mounted on glass slides. Slides were incubated in acetylthiocholine and copper-buffered solution at 40°C until tentacles appeared white. The stain was developed with a silver solution so that stained areas appear brown. Slides were incubated in the presence of the silver staining solution (+substrate) or saline (-substrate), rinsed according to protocol, and mounted in Fluoromount-G (SouthernBiotech) and imaged using a scanning, transmitted light microscope.

## Behavior
Discharge of nematocysts was assessed based on well-established assays (*Watson and Hessinger, 1994a*; *Gitter et al., 1994*). Adult *Nematostella* were placed in petri dishes containing a modified *Nematostella* medium, containing 16.6 mM $MgCl_2$. Animals were given appropriate time to acclimate before presented with stimuli. For assaying discharge, 5 mm round coverslips were coated with a solution of 25% gelatin (w/v) dissolved in medium, and allowed to cure overnight prior to use. Coverslips were presented to the animal's tentacles for 5 s and then immediately imaged at 20X magnification using a transmitted light source. To assay behavioral response to prey-derived chemicals, freshly hatched brine shrimp were flash frozen and pulverized, then filtered through a 0.22 µm filter. Coverslips were dipped in the prey extract and immediately presented to the animal. All pharmacological agents were bath applied, except for acetylcholine (1 mM), which was delivered as a bolus immediately prior to coverslip presentation. Acetylcholine exposure did not produce movement or contraction of tentacles. Experiments carried out in the absence of extracellular $Ca^{2+}$ were nominally $Ca^{2+}$ free and did not use extracellular chelators. The highest density of discharged nematocytes on the coverslip was imaged at 20X. Nematocyte discharge involves everting the barbed thread, causing them to embed in the gelatin-coated coverslips. Therefore, if nematocytes do not discharge, they are not captured by the gelatin-coated coverslip or visualized for quantification. Images were blindly analyzed using a custom Matlab routine (available in supplemental material) in which images were thresholded and the fraction of pixels corresponding to nematocytes was compared across experiments.

## Electron microscopy
Tentacles from an individual *Nematostella vectensis* were placed between two sapphire coverslips separated by a 100 µm spacer ring (Leica) and frozen in a high-pressure freezer (EM ICE, Leica). This

was followed by freeze-substitution (EM AFS2, Leica) in dry acetone containing 1% ddH2O, 1% OsO4 and 1% glutaraldehyde at −90°C for 48 hr. The temperature was then increased at 5°C/h up to 20°C and samples were washed at room temperature in pure acetone 3 × 10 min RT and propylene oxide 1 × 10 min. Samples were infiltrated with 1:1 Epon:propylene oxide overnight at 4°C. The samples were subsequently embedded in TAAB Epon (Marivac Canada Inc) and polymerized at 60°C for 48 hr. Ultrathin sections (about 50 nm) were cut on an ultramicrotome (Leica EM UC6) and collected with an automated tape collector (ATUM *Kasthuri et al., 2015*). The sections were then post-stained with uranyl acetate and lead citrate prior to imaging with a scanning electron microscope (Zeiss SIGMA) using a back-scattered electron detector and a pixel size of 4 nm.

Once all the sections were scanned, images were aligned into a stack using the algorithm 'Linear Stack Alignment with SIFT' available in Fiji (*Schindelin et al., 2012*). After alignment, images were imported into VAST (*Berger et al., 2018*) so that every cell could be manually traced. By examining sections and following cellular processes contacts between cells of interest (e.g. neurons and nematocytes) were identified and assessed for the presence of dense-core vesicles in the vicinity (~500 nm). Such instances were labeled as putative synapses. Cells were then rendered in three dimensions using 3D Studio Max 2019 (Autodesk, San Rafael, CA).

Nematocytes were readily identified because resin does not infiltrate the nematocyst capsule, making for an 'empty' appearance (large white area). Spirocytes were also readily identified based on their capsule containing a long, coiled filament. Sensory neurons were identified according to the higher number of dense core vesicles, higher number of synapses and the presence of sensory processes extending into the external environment.

## Statistical analysis

Data were analyzed with Clampfit (Axon Instruments) or Prism (Graphpad) and are represented as mean ± s.e.m. *n* represents independent experiments for the number of cells/patches or behavioral trials. Data were considered significant if $p < 0.05$ using paired or unpaired two-tailed Student's t-tests or one- or two-way ANOVAs. All significance tests were justified considering the experimental design and we assumed normal distribution and variance, as is common for similar experiments. Sample sizes were chosen based on the number of independent experiments required for statistical significance and technical feasibility.

## Acknowledgements

We thank J Lichtman and F Engert for helpful suggestions throughout this study, B Bean, D Julius, and R Nicoll for critical reading of the manuscript, K Boit for assistance with electron microscopy analyses, J Turecek for assistance with immunohistochemistry, and K Koenig for help with sea anemones. This research was supported by grants to NWB from the New York Stem Cell Foundation, Searle Scholars Program, Alfred P Sloan Foundation, Klingenstein-Simons Fellowship, and the NIH (R00DK115879), as well as the NIH (1F31NS117055) to KW, Swiss National Science Foundation (P2SKP3-187684) to CD and (P400PB-180894) LVG, and NIH (U24NS109102) to J Lichtman.

## Additional information

### Funding

| Funder | Grant reference number | Author |
| --- | --- | --- |
| New York Stem Cell Foundation | Robertson Neuroscience Investigator | Nicholas W Bellono |
| Chicago Community Trust | Searle Scholars Program | Nicholas W Bellono |
| Esther A. and Joseph Klingenstein Fund | Klingenstein-Simons Fellowship | Nicholas W Bellono |
| National Institutes of Health | R00DK115879 | Nicholas W Bellono |
| National Institutes of Health | 1F31NS117055 | Keiko Weir |
| Swiss National Science Foundation | P2SKP3-187684 | Christophe Dupre |

| National Institutes of Health | P2SKP3_187684 | Christophe Dupre |
| Swiss National Science Foundation | P400PB-180894 | Lena van Giesen |
| Alfred P. Sloan Foundation | | Nicholas W Bellono |
| Chicago Community Trust | Searle Scholars Program | Amy S-Y Lee |

The funders had no role in study design, data collection and interpretation, or the decision to submit the work for publication.

### Author contributions

Keiko Weir, Conceptualization, Data curation, Formal analysis, Investigation, Methodology, Writing - original draft, Writing - review and editing; Christophe Dupre, Data curation, Formal analysis, Methodology, Writing - review and editing; Lena van Giesen, Formal analysis, Writing - review and editing; Amy S-Y Lee, Data curation, Formal analysis, Writing - review and editing; Nicholas W Bellono, Conceptualization, Data curation, Formal analysis, Supervision, Funding acquisition, Investigation, Methodology, Writing - original draft, Project administration, Writing - review and editing

### Author ORCIDs

Keiko Weir https://orcid.org/0000-0002-2501-9352
Christophe Dupre http://orcid.org/0000-0002-5929-8492
Amy S-Y Lee https://orcid.org/0000-0002-4121-0720
Nicholas W Bellono https://orcid.org/0000-0002-0829-9436

### Decision letter and Author response

Decision letter https://doi.org/10.7554/eLife.57578.sa1
Author response https://doi.org/10.7554/eLife.57578.sa2

## Additional files

### Supplementary files

• Transparent reporting form

### Data availability

Deep sequencing data are available via the Sequence Read Archive (SRA) repository under the accession code PRJNA627705 and GenBank accession numbers are: cacna2d1 - MT334780, cacna1a - MT334781, cacnb2 - MT334782, nompC - MT334783. Other data are provided in associated source data files.

The following datasets were generated:

| Author(s) | Year | Dataset title | Dataset URL | Database and Identifier |
| --- | --- | --- | --- | --- |
| Weir K, Dupre C, van Giesen L, Lee ASY, Bellono NW | 2020 | Nematostella vectensis RNA sequencing | https://www.ncbi.nlm.nih.gov/sra/PRJNA627705 | NCBI Sequence Read Archive, PRJNA627705 |
| Weir K, Dupre C, van Giesen L, Lee ASY, Bellono NW | 2020 | Nematostella vectensis cacna2d1 mRNA, complete cds | https://www.ncbi.nlm.nih.gov/nucleotide/MT334780 | NCBI GenBank, MT334780 |
| Weir K, Dupre C, van Giesen L, Lee ASY, Bellono NW | 2020 | Nematostella vectensis cacna1a mRNA, complete cds | https://www.ncbi.nlm.nih.gov/nucleotide/MT334781 | NCBI GenBank, MT334781 |
| Weir K, Dupre C, van Giesen L, Lee ASY, Bellono NW | 2020 | Nematostella vectensis cacnb2 mRNA, complete cds | https://www.ncbi.nlm.nih.gov/nucleotide/MT334782 | NCBI GenBank, MT334782 |
| Weir K, Dupre C, van Giesen L, Lee ASY, Bellono NW | 2020 | Nematostella vectensis nompC mRNA, complete cds | https://www.ncbi.nlm.nih.gov/nucleotide/MT334783 | NCBI GenBank, MT334783 |

The following previously published dataset was used:

| Author(s) | Year | Dataset title | Dataset URL | Database and Identifier |
|---|---|---|---|---|
| Sunagar K, Columbus-Shenkar YY, Fridrich A, Gutkovich N, Aharoni R, Moran Y | 2018 | Nematostella vectensis Raw sequence reads | https://www.ncbi.nlm.nih.gov/bioproject/PRJNA391807/ | NCBI BioProject, PRJNA391807 |

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
