## [Decision Letter]

**Acceptance summary:**

This is an elegant study of an interesting biological system, nematocysts that harbor the stinging, venom-infused harpoons used by jellyfish and anemones. The study is comprehensive with cell biology, ion channel biophysics, transcriptomics, and ultrastructural analysis embedded in a neuroethological context. Especially intriguing are the interactions of mechanosensory and chemical input to trigger the discharge and the role of an unusual calcium channel inactivation and the relief of this inactivation by synaptic input to the nematocyst that triggers a long duration action potential and ejection of the nematocyst's harpoon.

**Decision letter after peer review:**

Thank you for submitting your article "A molecular filter for the cnidarian stinging response" for consideration by *eLife*. Your article has been reviewed by three peer reviewers, and the evaluation has been overseen by Ronald Calabrese as the Senior and Reviewing Editor. The following individuals involved in review of your submission have agreed to reveal their identity: Harold H Zakon (Reviewer #1); Sviatoslav N Bagriantsev (Reviewer #2).

The reviewers have discussed the reviews with one another and the Reviewing Editor has drafted this decision to help you prepare a revised submission.

Summary:

This is an elegant study of an interesting biological system. While most people know that jellyfish sting their prey (or unwary swimmers), most do not know about the complex cells called nematocysts that harbor the stinging, venom-infused harpoons used by jellyfish and anemones. This study takes a soup to nuts approach with cell biology, ion channel biophysics, transcriptomics, and ultrastructural analysis embedded in a neuroethological context. Further, the ion channel biophysics ranges from calcium and calcium-activated K channels to mechanoreceptors to neurotransmitter receptors, calling on a variety of usually distinct topics blended in a comprehensive narrative. Especially intriguing are the interactions of mechanosensory and chemical input to trigger the discharge and the role of an unusual calcium channel inactivation and the relief of this inactivation by synaptic input to the nematocyst that triggers a long duration action potential and ejection of the nematocyst's harpoon.

Essential revisions:

This paper was praised by the reviewers for its thoroughness and creativity. There are suggested revisions that will make this strong paper stronger contained in the reviewer's comments. Revisions will be handled by the Reviewing Editor and further review is not necessary. Reviewer #3 indicates some further experiments; these need not be pursued but if already completed could be included. Discussion should acknowledge that these experiments would further support the conclusions. Further amplifying the text as suggested by reviewer #2 could broaden the paper's appeal to the general reader.

Reviewer #1:

I had only one question. The nCa_V_ channels are inactivated in the range of normal resting potentials (for these cells, -64 mV). Figure 1D shows that nCa_v_ need membrane potentials below ~ -80 mV to recover. In the voltage clamp analysis membrane potentials were held at -110 mV for complete recovery from inactivation. I note that nematocysts will spike with membrane potential moved to -90 mV, presumably when moved there by a calcium-dependent K^+^ current. Perhaps I missed this but what is the reversal potential for the K^+^ current? Unless it reverses at ~ -80 or more negative, will there be enough hyperpolarization to remove the nCa_v_ inactivation?

Reviewer #2:

My main textual criticism is that the manuscript is written very densely. Given that the *eLife* format is not size-limiting, I'd suggest to expand on experimental details in the text whenever possible. Otherwise, the manuscript is going to be largely accessible to electrophysiologists, even though its scope and interest goes well beyond the biophysics community.

Figure 1F – If I'm not mistaken, here authors recorded either in their standard solutions (Na), or substituted extracellular Na with NMDG. But is not at all clear from the text/legend. Please clarify.

Figure 1G and Figure 5D – I'm struggling to understand why the images with non-discharged nematocytes look like they don't have anything at all, whereas I expect to see intact nematocytes as in Figure 1B. At the same time, the discharged images look like they have intact nematocytes. Please add some visual guides/explanation.

Subsection “Nematocyte Ca_V_ channels”, and elsewhere – "activation threshold". I'd suggest to use "apparent activation threshold" throughout.

Figure 2C, D. Please indicate that the lines above the plots refer to 5-s holding voltages between test pulses at 0mV. As presented, it is very unclear what was done without reading the Materials and methods section.

Reviewer #3:

1) Are there homologs of Piezo1/2 in these animals? If so, there are activators and inhibitors the authors could test. (The authors should not call Gd^3+^ "a mechanosensitive receptor blocker". This implies specificity, but it blocks many divalent permeant ion channels).

2) The authors do not specify what they mean by 'chemical signals' from prey, or 'prey-derived extract'. They should describe more clearly what these might be, and describe in the text and Materials and methods section how they isolated it and what was tested.

3) It is unclear if ACh is secreted from spirocytes or neurons, and whether the EM showed vesicle-containing junctions with either. If the source of ACh release could be determined, it would add an important new point to the paper.

---

## [Author Response]

Reviewer #1:I had only one question. The nCa_v_ channels are inactivated in the range of normal resting potentials (for these cells, -64 mV). Figure 1D shows that nCa_v_ need membrane potentials below ~ -80 mV to recover. In the voltage clamp analysis membrane potentials were held at -110 mV for complete recovery from inactivation. I note that nematocysts will spike with membrane potential moved to -90 mV, presumably when moved there by a calcium-dependent K^+^ current. Perhaps I missed this but what is the reversal potential for the K^+^ current? Unless it reverses at ~ -80 or more negative, will there be enough hyperpolarization to remove the nCa_v_ inactivation?

Under our experimental conditions, the estimated K^+^ reversal potential is approximately -100mV. This is negative to the resting membrane potential and consistent with K^+^-mediated hyperpolarization to relieve nCa_V_ inactivation. We now include this value in the text.

Reviewer #2:My main textual criticism is that the manuscript is written very densely. Given that the eLife format is not size-limiting, I'd suggest to expand on experimental details in the text whenever possible. Otherwise, the manuscript is going to be largely accessible to electrophysiologists, even though its scope and interest goes well beyond the biophysics community.

We modified the text in several areas to explain electrophysiological findings, add context, and/or reduce technical terminology, which is further described in the legends and in the Materials and methods section.

Figure 1F – If I'm not mistaken, here authors recorded either in their standard solutions (Na), or substituted extracellular Na with NMDG. But is not at all clear from the text/legend. Please clarify.

Indeed, substitution of Na^+^ and Ca^2+^ was carried out to determine which major cation contributed to I_CaV_. As described, Na^+^ had little effect compared with Ca^2+^. We have modified the legend for clarity.

Figure 1G and Figure 5D – I'm struggling to understand why the images with non-discharged nematocytes look like they don't have anything at all, whereas I expect to see intact nematocytes as in Figure 1B. At the same time, the discharged images look like they have intact nematocytes. Please add some visual guides/explanation.

Nematocyte discharge involves everting the barbed thread, causing them to embed in the gelatin-coated coverslips used in our assay for detecting discharged nematocytes. If nematocytes do not discharge, they are not captured by the gelatin-coated coverslip or visualized for quantification. Discharged nematocysts are frequently embedded perpendicular to the coverslip surface so that the cyst (the majority of the nematocyte cell, so similar in appearance) is the most visible component versus the discharged thread. We have modified the legend in Figure 1 for clarity and added further explanation to the Materials and methods section.

Subsection “Nematocyte Ca_V_ channels”, and elsewhere – "activation threshold". I'd suggest to use "apparent activation threshold" throughout.

“Activation threshold” has been changed to “apparent activation threshold” throughout.

Figure 2C, D. Please indicate that the lines above the plots refer to 5-s holding voltages between test pulses at 0mV. As presented, it is very unclear what was done without reading the Materials and methods section.

We modified the legend for clarity regarding the voltage protocol.

Reviewer #3:1) Are there homologs of Piezo1/2 in these animals? If so, there are activators and inhibitors the authors could test. (The authors should not call Gd^3+^ "a mechanosensitive receptor blocker". This implies specificity, but it blocks many divalent permeant ion channels).

It is possible that *Nematostella vectensis* expresses Piezo1/2 channels, but we did not detect enriched homologous transcripts and it is not clear that specific agonists or antagonists exist for these uncharacterized channels. Our data support that NompC may contribute to nematocyte mechanotransduction but we acknowledge other described or uncharacterized mechanoreceptors could be involved. Therefore, we convey a hypothetical tone regarding molecular identification of the nematocyte mechanoreceptor(s) and emphasize that our main conclusion regards the intrinsic cellular mechanosensitvity. We added text to specify that Gd^3+^ also blocks other cation channels.

2) The authors do not specify what they mean by 'chemical signals' from prey, or 'prey-derived extract'. They should describe more clearly what these might be, and describe in the text and Materials and methods section how they isolated it and what was tested.

We further specified the methodology for isolating prey extract and discussed the potential active molecules. We completely agree that identification of the chemosensory cell would be a great future direction toward further understanding sensory integration, chemical coding, and synaptic regulation. We now further expand upon this in the Discussion section.

3) It is unclear if ACh is secreted from spirocytes or neurons, and whether the EM showed vesicle-containing junctions with either. If the source of ACh release could be determined, it would add an important new point to the paper.

As noted above, we agree that identification of chemosensory cells and the source of transmitter(s) that regulate nematocytes would be a fantastic future direction. We have expanded upon this in the Discussion section.